# Isoalantolactone (IAL) Regulates Neuro-Inflammation and Neuronal Apoptosis to Curb Pathology of Parkinson’s Disease

**DOI:** 10.3390/cells11182927

**Published:** 2022-09-19

**Authors:** Dewei He, Yanting Liu, Jie Li, Hefei Wang, Bojian Ye, Yuan He, Zhe Li, Xiyu Gao, Shoupeng Fu, Dianfeng Liu

**Affiliations:** 1College of Animal Science, Jilin University, Changchun 130012, China; 2Department of Neurosurgery, Seoul St. Mary’s Hospital, College of Medicine, Catholic University of Korea, Seoul 296-12, Korea; 3College of Veterinary Medicine, Jilin University, Changchun 130012, China

**Keywords:** isoalantolactone, PD, neuro-protection, neuro-inflammation, apoptosis

## Abstract

Parkinson’s disease (PD) is a neurodegenerative disease in which neuronal apoptosis and associated inflammation are involved in its pathogenesis. However, there is still no specific treatment that can stop PD progression. Isoalantolactone (IAL) plays a role in many inflammation-related diseases. However, its effect and mechanism in PD remain unclear. In this study, results showed that IAL administration ameliorated 1-methyl-4-phenyl-1, 2, 3, 6-tetrahydropyridine (MPTP)-induced PD-related pathological impairment and decreased motor activity in mice. Results from in vitro mechanistic studies showed that IAL regulated apoptosis-related proteins by activating the AKT/Nrf2 pathway, thereby suppressing the apoptosis of SN4741 cells induced by N-methyl-4-phenylpyridinium Iodide (MPP^+^). On the other hand, IAL inhibited LPS-induced release of pro-inflammatory mediators in BV2 cells by activating the AKT/Nrf2/HO-1 pathway and inhibiting the NF-κB pathway. In addition, IAL protected SN4741 from microglial activation-mediated neurotoxicity. Taken together, these results highlight the beneficial role of IAL as a novel therapy and potential PD drug due to its pharmacological profile.

## 1. Introduction

Parkinson’s disease (PD), a multifactorial disease closely related to age, is more common in middle-aged and elderly people. Most PD patients are sporadic cases, and few have a family history [1,2]. The main pathology of PD is dopaminergic neuron degeneration in the substantia nigra (SN) and a marked DA reduction in the striatum [3,4]. At present, the exact mechanism of dopaminergic neuron loss is still unclear, and there is no effective treatment to avoid dopaminergic neuron damage. Clinically, the discovery of PD mainly depends on medical history, clinical symptoms and signs. PD has a long course and insidious onset and is only diagnosed when 50% of neurons die, so PD treatment generally starts late [5,6]. For severe patients, there is currently no cure. Therefore, it is necessary to deeply study the pathogenesis of PD and find better treatment methods.

Apoptosis is a death process that cells in the body actively strive for to better adapt to the living environment. It is the process whereby the body clears away useless or harmful cells. Strictly speaking, apoptosis is a beneficial result of gene regulation. However, in the disease state, apoptosis is prone to disorder, which damages normal cells and aggravates the disease [7,8,9]. Studies have shown that there are a large number of apoptotic neurons in the midbrain of PD patients. During PD, causative factors lead to apoptosis of healthy neurons, and apoptotic signaling in turn leads to additional pathological risks, such as neuro-inflammation. These pathological factors lead to further neuronal damage. Neuronal damage leads to decreased release of striatal dopamine (DA) and its metabolites (DOPAC and HVA), which in turn leads to decreased motor activity, and severe cases show dyskinesia [10,11,12]. Therefore, inhibiting excessive neuronal apoptosis and finding neuroprotective drugs is an important direction for the treatment of PD. At the same time, inhibiting neuro-inflammation reduces neuronal damage to a certain extent and slows down the process of PD.

Isoalantolactone (IAL), a sesquiterpene lactone compound extracted from chicory root, is naturally present in a variety of vegetables and plants. IAL is widely used in Chinese herbal medicine due to its unique pharmacological effects. Studies have reported that IAL exerts anti-inflammatory and anti-tumor effects in peripheral tissues [13,14]. Song et al. found that IAL alleviates ovalbumin-induced asthmatic inflammation by reducing alternately activated macrophages and STAT6/PPAR-gamma/KLF4 signaling [15]. He et al. found that IAL inhibited LPS-induced inflammation by activating NF-κB in macrophages and improved survival in sepsis mice [16]. Yuan and Ding et al. found that IAL suppressed LPS-induced inflammation and alleviated acute lung injury by inhibiting TRAF6 ubiquitination and activating Nrf2 [17,18]. In addition, IAL also plays an important role in neurological diseases. Seo et al. found that IAL prevents amyloid β 25–35-induced toxicity of cortical neurons in mice and scopolamine-induced cognitive impairment in mice [19]. Wang et al. found that IAL inhibited inflammation by activating the GSK-3β-Nrf2 signaling pathway in the BV2 cell [20]. In sum, IAL plays a role in numerous inflammation-related diseases. However, its effect and mechanism in PD remain unclear. Therefore, this experiment aimed to investigate the effect of IAL on neuro-inflammation and neuronal apoptosis to explore its potential in PD therapy.

## 2. Materials and Methods

### 2.1. Reagents

Isoalantolactone (IAL) was obtained from Chengdu PuFeiDe Biotech Co., Ltd. (Chengdu, China). 1-Methyl-4-phenyl-1, 2, 3, 6-tetrahydropyridine (MPTP), lipopolysaccharide (LPS), dimethyl sulfoxide (DMSO) and N-methyl-4-phenylpyridinium iodide (MPP^+^ iodide) were purchased from Sigma Aldrich (St Louis, MO, USA). RA (nrf2 inhibitor) and MK2206 (AKT inhibitor) were obtained from Selleck Chemicals Aldrich (Shanghai, China). Dulbecco’s modified eagle’s medium (DMEM), trypsin (0.05% and 0.25%) and serum (FBS) were obtained from Gibco (Grand Island, NY, USA).

### 2.2. Animals and Models

Thirty male C57/BL6 mice (10 weeks old) were obtained from Liaoning Chang Sheng (Liaoning, China). Mice were given free access to water and food. Mice were housed in an environment with a 12 h/12 h light/dark cycle. Before the experiment, mice were randomly divided into three groups: a blank control group (intraperitoneally injected with saline and gavaged with 2% carboxy methylcellulose sodium with DMSO), a single-injection MPTP group (intraperitoneally injected MPTP and gavaged with 2% carboxy methylcellulose sodium with DMSO) and an IAL + MPTP group (intraperitoneally injected MPTP and gavaged with IAL). The experimental process is shown in Figure 1A. In short, MPTP dissolved in saline was injected intraperitoneally into mice at a dose of 30 mg/kg/d for 7 days. Behavioral testing of mice was performed one week after MPTP injection. During the experiment, IAL (10 mg/kg dissolved in DMSO and 2% carboxy methylcellulose sodium) was administered by gavage for 17 days. This experiment was approved by the Institutional Animal Care and Use Committee of Jilin University (Changchun, China, protocol number SY202107004).

### 2.3. Behavioral Testing

After MPTP injection for 1 week, behavioral testing was performed with reference to previous studies [21,22,23,24].

#### 2.3.1. Open Field Test

Before the test, the mice were put into the box 3 days in advance to acclimatize them to the environment. During the experiment, a quiet and noise-free environment was maintained. The space of the open field box was 50 cm × 50 cm × 30 cm. During the experiment, the mice were placed in the open field and allowed to move freely. The total distance the mice travelled in 5 min and their time in the central zone were tracked and recorded using video cameras. The travel trajectories of mice were presented using Any-maze software (Stoelting Co., Chicago, IL, USA). The experimental setup and operation followed the manufacturer’s instructions.

#### 2.3.2. Climbing Pole Test

Before the test, mice were trained for this experiment for 3 days. The wooden pole used in the experiment was 50 cm long, and a ball was fixed on the top. At the beginning of the test, the mice were placed on the ball. When the ball climbed to the stick, a stopwatch was used to record the time at this moment as A; when it fell to the bottom of the stick, the time at the moment was recorded as B. Thus, the time it took for the mouse to climb the entire stick was C, with C = A − B. Each test was repeated three times and the mean value was calculated.

#### 2.3.3. Rotating Rod Test

Before the test, mice were trained on a rotating rod (diameter 30 cm, length 60 cm) for 3 days to get used to the experiment. During the experiment, the mice were placed on the rotarod in the center of the roller. Once the rpm was set and the power was turned on, the wheels turned automatically, and the turning stick gradually increased the rotational speed from 10 rpm to 50 rpm over a period of 5 min. Mice were allowed to acclimatize once and then performed three consecutive trials of 5 min each. The rest period between each trial was 30 min. The residence time of the mice on the rotating rod was recorded. Each test was repeated three times and the mean value was calculated.

### 2.4. Immunohistochemistry (IHC) Staining

After modeling, mice were sacrificed by euthanizing them and the mice’s midbrains were soaked in 4% paraformaldehyde for 36 h. Subsequently, midbrain tissues were placed in graded alcohol (70%, 80%, 90%, 95% and 100 % each for 1 h) for dehydration. After dehydration was completed, the tissues were placed in xylene to clear them for 15 min and then in dipping wax for 60 min. Then, the tissues were sliced into sections of 6 µm each and dried at 75 °C. After that, the IHC staining was performed using an Ultrasensitive TMS-P kit (Biological Technology, Wuhan, China) according to the manufacturer’s protocols. The dopaminergic neurons were marked with the anti-tyrosine hydroxylase (TH) antibody (1:500, Santa, Shanghai, China) and microglia were marked with the anti-ionized calcium-binding adapter molecule 1 (IBA-1) antibody (1:100, Santa, Shanghai, China). Results were analyzed using Image J software.

### 2.5. High-Performance Liquid Chromatography (HPLC) Assay

After modeling, mice were sacrificed by euthanizing them, and the mice’s striata were obtained and HPLC experiments performed to measure dopamine (DA) and its metabolites (DOPAC and HVA) in the striata. Briefly, 0.2 g of striatal tissue was placed in an EP tube, followed by the addition of 1 mL perchloric acid (1 M). After that, the tissues were thoroughly ground in a grinder. Subsequently, the sample was centrifuged at 12,000 rpm for 15 min. After centrifugation, the supernatant was transferred to a new tube. One-half volume of a potassium dihydrogen phosphate solution was added to the supernatant and centrifuged at 12,000 rpm for 10 min at 4 °C. Then, 20 μL of the sample was injected into an HPLC system for analysis.

### 2.6. Cell Culture

SN4741 cells (mouse dopaminergic neuron) and BV-2 cells (mouse microglia) were purchased from Shanghai Bin Sui Biotechnology (Shanghai, China). The cells were cultured under the following conditions: 90% DMEM + 10% FBS; air atmosphere, 95%; carbon dioxide (CO2), 5%; temperature, 37°C. When the density reached 80%, the cells were digested and passaged with 0.25% (SN4741 cells) and 0.05% (BV-2 cells) trypsin.

### 2.7. CCK-8 Assay

Cells were seeded into 96-well plates at a density of 2–3 × 10^4^ per well. When the density reached 50%, IAL and DMSO were added into the wells and cells were cultured for another 24 h. After that, the medium was replaced with CCK-8 dilution (Shang Bao Biological, Shanghai, China). Subsequently, cells were cultured for another 3 h and then absorbance was measured with a microplate reader (450 nm).

### 2.8. ELISA Assay

Cells were seeded in 24-well plates. When the cells were around 70% grown, the medium was replaced with fresh DMEM. Then, cells were treated with different stimuli (IAL and LPS). Subsequently, the expression of IL-6 and TNF-α in the supernatant was measured with ELISA kits (Bio Legend, San Diego, CA, USA) according to the manufacturer’s protocol.

### 2.9. LDH Assay

Cells were seeded into 96-well plates at a density of 2–3 × 10^4^ per well. When the cells grew to about 70%, different stimuli (IAL, MPP^+^ or conditioned medium) were added to the culture wells. After 24 h, the level of LDH released in the medium was determined using an LDH assay kit (Beyotime Inst Biotech, Beijing, China) according to the manufacturer’s protocol instructions.

### 2.10. RT-PCR Assay

RNA of cells and midbrain tissues was extracted using trizol (Sigma Aldrich St Loui, MO, USA), chloroform and isopropanol (Shanghai Aladdin Biochemical Technology Co., Ltd., Shanghai, China). Then, 0.5 μg of RNA was reverse transcribed into cDNA using a reverse transcription kit (Invitrogen, Carlsbad, CA, USA). Subsequently, cDNA was amplified using the 2 × M5 Hiper Real Time PCR Super mix (Poly Mei Biotechnology Co., Ltd., Beijing, China) and Cq values were recorded with a Bio Rad system. The mRNA of the mediators was assessed relative to β-actin according to the 2^−ΔΔCT^.

The primers are listed in Table 1.

### 2.11. Western Blot Assay

Total proteins of midbrain tissues and cells were extracted using protein extraction kits (Beyotime Biotechnology, Shanghai, China). The extracted proteins were stored in a −80 °C refrigerator after packaging. In the experiment, electrophoresis (110 V, 2 h) was performed with 30 μg proteins. After electrophoresis, proteins were transferred to methanol (Shanghai Aladdin Biochemical Technology Co., Ltd., Shanghai, China)-treated fiber PVDF membranes (Millipore, Billerica, MA, USA) (90 V, 1.5 h). The membranes were then blocked for 4 h in 5% skim milk. Subsequently, the membranes were incubated with the primary antibody (p-NF-κB p65 (1:500), AKT (1:2000), p-AKT (1:500), Nrf2 (1:1000), PCNA (1:4000), iNOS (1:2000), COX2 (1:2000), IκB (1:500), p-IκB (1:500), Bax (1:2000), Bcl-2 (1:2000), caspase 3 (1:500), NF-κB p65 (1:1000), β-actin (1:4000)) at 4 °C for 12 h and the secondary antibody (1:5000) at room temperature for 1 h. The primary antibodies were obtained from Proteintech (Wuhan, China). The secondary antibody was obtained from Invitrogen (Waltham, CA, USA). After that, proteins were labeled using the Super ECL Detection Reagent (Yeasen Biotechnology, Shanghai, China) and imaged using Kodak X-ray film (Ruike Medical Equipment Co., Ltd., Xiamen, China). Results were analyzed using ImageJ.

### 2.12. Flow Cytometry

Cells were seeded into six-well plates at a density of 2–3 × 10^5^ per well. When the cells grew to about 70%, different stimuli (IAL or MPP^+^) were added into wells. After 24 h, the apoptosis levels of cells were detected using Annexin V-FITC/PI detection kits (Beyotime Biotechnology, Shanghai, China) according to the manufacturer’s protocol.

### 2.13. Tunel Staining

Cells were seeded into 24-well plates with slides (9 mm × 9 mm) on the bottom at a density of 2–3 × 10^4^ per well. When the cells grew to about 70%, the conditioned medium was added to the wells. After 24 h, medium was removed and the apoptosis levels of the cells were detected using Tunel cell death detection kits (Roche, Shanghai, China) according to the manufacturer’s protocol.

### 2.14. Data Analyses

Data differences were analyzed using SPSS 19.0 software (IBM). All data were presented as means ± SEM. One-way analysis of variance combined with multiple comparisons was used to compare the differences between different treatment groups. A *p*-value of <0.05 was considered to be statistically significant.

## 3. Results

### 3.1. IAL Treatment Alleviates the Mobility of MPTP-Induced PD Mice

Clinically, PD patients present with motor dysfunction. To explore the effect of IAL on PD, we first investigated the effect of IAL on motor activity in MPTP-injected mice via a behavioral test (Figure 1A). In the open field experiment, MPTP treatment resulted in a decrease in the motor activity of the mice, mainly manifested as a decrease in the total distance traveled in the zone and a decrease in the ability to explore the central zone. IAL administration alleviated this situation (Figure 1B–D). In the pole-climbing experiment, MPTP treatment resulted in decreased limb coordination in mice, which was manifested as a longer time required for pole-climbing. IAL treatment relieved this symptom (Figure 1E). In the rotarod experiment, MPTP treatment led to a decrease in the fatigue resistance of mice, which was manifested as a decrease in the duration on the rotarod fatigue apparatus. IAL treatment relieved this symptom (Figure 1F). These results prove that IAL alleviates the mobility of MPTP-induced PD mice.

### 3.2. IAL Treatment Decreases Dopaminergic Neuron Degeneration and Inhibits Microglia Over-Activation in MPTP-Induced PD Mice

The main pathology of PD is the loss of dopaminergic neurons in the SN. To explore the effect of IAL on PD, we investigated the effect of IAL on dopaminergic neuron damage in MPTP-injected mice. The results showed that IAL significantly inhibited MPTP-induced dopaminergic neuron damage (Figure 2A,B). In PD, the activation of a large number of microglia occurs simultaneously with neuronal injury. Therefore, we further investigated the effect of IAL on microglial activation. The results showed that IAL significantly inhibited the excessive activation of microglia (Figure 2A,C). Western blot results further confirmed the above findings at the protein level (Figure 2D–F). These results prove that IAL decreases dopaminergic neuron degeneration and inhibits microglia activation in MPTP-injected PD mice.

### 3.3. IAL Treatment Suppresses Midbrain Inflammatory Responses and Increases Striatal Dopamine and Its Metabolite Levels in MPTP-Induced PD Mice

To further elucidate the effect of IAL on PD, we investigated the effect of IAL on midbrain inflammatory response and striatal DA and its metabolites (DOPAC and HVA) in mice. The ELISA results showed that IAL treatment significantly suppressed the release of pro-inflammatory mediators, such as IL-6, IL-1β and TNF-α, in midbrain tissue (Figure 3A–C). The HPLC results showed that IAL increased levels of striatal DA and its metabolite HVA but not DOPAC (Figure 3D).

### 3.4. IAL Treatment Reduces Cells Death in MPP^+^-Exposed SN4741 Cells

The main pathology of PD is the death of midbrain dopaminergic neurons. We cultured mouse dopaminergic neuron cell lines in vitro to study the effect of IAL on MPP^+^-induced neuronal damage. Flow cytometry results showed that IAL (2 μM) inhibited the death of SN4741 cells induced by MPP^+^ (Figure 4A,B). The CCK-8 results showed that IAL ameliorated the decrease in SN4741 cell viability caused by MPP^+^ (Figure 4C). The LDH results showed that IAL reduced MPP^+^-induced LDH release in SN4741 cells (Figure 4D).

Subsequently, we examined the effect of IAL on caspase 3, Bax and Bcl-2 in MPP^+^-exposed SN4741 cells. Results showed that IAL suppressed expression of apoptotic proteins caspase 3 and Bax and increased expression of anti-apoptotic protein Bcl-2 (Figure 4E–H).

### 3.5. IAL Treatment Reduces MPP^+^-Induced Apoptosis of SN4741 Cells by Activating the AKT/Nrf2 Signaling Pathway

Studies have shown that the Nrf2 and AKT pathways are related to cell damage. Therefore, we further investigated the effect of IAL (2 μM) on the Nrf2 and AKT pathways of SN4741 cells. Results showed that IAL activated the AKT and Nrf2 pathways in SN4741 cells (Figure 5A–C).

Our further investigations found that inhibitors of AKT and Nrf2 reversed the regulation of IAL in apoptotic proteins (Figure 5D–G). Taken together, these data prove that IAL reduces MPP^+^- induced apoptosis of SN4741 cells by activating the AKT/Nrf2 signaling pathway.

### 3.6. IAL Treatment Suppresses the Release of Pro-Inflammatory Mediators in LPS-Induced BV2 Cells

Microglia are the main effector cells of neuro-inflammation. To elucidate the mechanism by which IAL affects PD, we next examined the effect of IAL on microglial inflammation. First, we examined the effect of different concentrations of IAL (0.5 μM, 1 μM, 2 μM, 4 μM, 8 μM) on the growth of BV2 cells. It was found that up to 2 μM of IAL had no significant effect on BV2 (Figure 6A). Then, we incubated cells with IAL (1 μM and 2 μM) and LPS (1 μg/mL) for 12 h, extracted cellular RNA for reverse transcription and detected the mRNA expression of pro-inflammatory mediators using the quantification PCR method. The results showed that IAL significantly suppressed the mRNA expression of pro-inflammatory mediators (IL-6 (Figure 6B), TNF-α (Figure 6C), iNOS (Figure 6D), COX2 (Figure 6E)) in BV2 cells. In addition, we also checked the protein expression of the mediators using Western blot and ELISA methods. The results showed that IAL treatment inhibited the protein expression of pro-inflammatory mediators (IL-6 (Figure 6F), TNF-α (Figure 6G), iNOS (Figure 6H,I), COX2 (Figure 6H,J)) in BV2 cells.

### 3.7. IAL Treatment Inhibits Neuro-Inflammation by Activating the AKT/Nrf2/HO-1 Pathway and Suppressing the NF-κB Signaling Pathway

To elucidate the mechanism by which IAL suppresses the release of pro-inflammatory mediators, we investigated the effect of IAL on inflammation-related pathways. Firstly, we incubated BV2 cells with IAL (2 μM) for different times, and then collected cells to detect the activation of the AKT, Nrf2 and HO-1 pathways. Results showed that IAL treatment significantly activated the AKT/Nrf2/HO-1 signaling pathway (Figure 7A–D). We then incubated BV2 cells with IAL (1 μM, 2 μM) and LPS (1 μg/mL) for 1 h, collected cells to extract total protein and detected activation of the NF-κB pathway. Results displayed that IAL treatment inhibited activation of NF-κB (Figure 7E–H).

Subsequently, we treated BV2 cells with MK2206 (10 μM, AKT inhibitor), RA (5 μM, Nrf2 inhibitor) and SnPP (40 μM, HO-1 inhibitor), respectively, and then investigated the effect of IAL on the release of inflammatory mediators. The results showed that MK2206, RA and SnPP treatment reversed the inhibitory effect of IAL on pro-inflammatory mediators to varying degrees (Figure 7I–L).

Taken together, these results prove that IAL treatment inhibits neuro-inflammation by activating the AKT/Nrf2/HO-1 pathway and suppressing the NF-κB signaling pathway.

### 3.8. IAL Treatment Protects SN4741 from Microglial Activation-Mediated Neurotoxicity

Studies have shown that inflammatory mediators released by activated microglia can damage neurons. Our previous study found that IAL inhibited LPS-induced microglial inflammation and MPP^+^-induced SN4741 cell damage. To further elucidate the neuroprotective effect of IAL, we cultured SN4741 cells with conditioned medium made from the culture supernatant of activated microglia. Then, the damage of neurons was detected with the CCK-8, LDH and Tunel staining methods (Figure 8A). The results showed that IAL increased neuronal cell survival (Figure 8B) and decreased LDH release (Figure 8C) and Tunel staining level in SN4741 cells (Figure 8D). Taken together, these data prove that IAL treatment protects SN4741 from microglial activation-mediated neurotoxicity.

## 4. Discussion

In this study, we confirmed that IAL ameliorates PD symptoms by inhibiting neuro-inflammation and neuronal apoptosis (Figure 9). Using in vivo studies, we found that IAL administration ameliorated MPTP-induced PD-related pathological damage and decreased motor activity in mice. In the in vitro mechanistic studies, IAL inhibited MPP^+^-induced neuronal apoptosis by activating the AKT/Nrf2 signaling pathway. On the other hand, IAL inhibited LPS-induced microglia inflammation by activating the AKT/Nrf2/HO-1 pathway and inhibiting the NF-κB pathway. In addition, we also found that IAL protects SN4741 from microglial activation-mediated neurotoxicity.

The main pathological change in PD is death of dopaminergic neurons in the SN, resulting in a decrease in striatal DA. The main clinical manifestations of PD patients are motor symptoms, such as myotonia, resting tremor and balance disorders [25,26]. MPTP, a neurotoxin, is metabolized after entering brain by MAOB produced by microglia into the toxic cation MPP^+^. MPP^+^ causes PD-like symptoms by destroying dopaminergic neurons in the SN. It is widely used in the study of various animal models of PD [27,28]. In the current experiment, we studied the effect of IAL on pathological damage and PD-like symptoms in MPTP-injected mice. It was found that IAL improved the number of dopaminergic neurons in the SN of MPTP-injected mice (Figure 2) and the level of striatal DA (Figure 3), thereby ameliorating their motor performance (Figure 1). However, the improvement effect of IAL on dopamine metabolite levels in the striatum was not evident (Figure 3D).

The exact cause of PD is still unknown. Genetic factors, environmental factors and aging may all be involved in the death of dopaminergic neurons [29,30]. Studies have shown that inflammation and innate immunity are involved in the progression of PD. Microglia are innate immune cells in the CNS. When PD occurs, a large number of microglia aggregate to the lesion site and are activated to initiate the innate immune response. However, the continuously activated microglia produce a large amount of pro-inflammatory mediators, such as IL-6 and TNF-α, which further cause neuronal damage and aggravate PD [31,32,33]. In this experiment, the results showed that the number of microglia was significantly increased in the SN of MPTP-injected mice, and IAL administration suppressed this phenomenon. In addition, the results also showed that IAL administration significantly reduced the level of pro-inflammatory factors in the midbrain and serum of MPTP-injected mice (Figure 2 and Figure 3).

Midbrain dopaminergic neurons are the main damaged cells in PD. Studies have shown that a major cause of PD is persistent damage to midbrain dopaminergic neurons [34,35]. SN4741, a mouse dopaminergic neuron cell line, is widely used in dopaminergic neuron-related research [36,37]. In this study, we found that IAL protected SN4741 cells from MPP^+^-induced damage and apoptosis (Figure 4). Normal apoptosis involves the autonomous and orderly death of cells controlled by genes. However, during PD, apoptosis is disrupted, the cells express more pro-apoptotic proteins (Bax and caspase 3) and less anti-apoptotic protein (Bcl-2) [38,39]. In this study, we found that IAL inhibited the expression of Bax and caspase 3 and promoted the expression of Bcl-2 (Figure 4). Our studies showed that, after entering cells, MPP^+^ causes oxidative stress by damaging mitochondria, leading to neuronal apoptosis [40,41]. AKT, an intracellular kinase, plays a vital role in cell survival and apoptosis. Nrf2 is a nuclear transcription factor, which is transferred to the nucleus after activation and binds to AREs, activates the transcription of downstream genes and then translates a series of related proteins to play physiological functions [42,43]. Studies have shown that AKT and Nrf2 are important regulatory pathways for oxidative stress [44,45]. In this study, we proved that IAL suppressed MPP^+^-induced SN4741 cell apoptosis by activating the AKT/Nrf2 pathway (Figure 5).

During PD, neuronal apoptosis is accompanied by inflammatory responses, and persistent inflammatory responses even exacerbate neuronal damage [31,46]. In this experiment, we investigated the effect of IAL on neuro-inflammatory responses. Our results showed that IAL inhibited LPS-induced production of pro-inflammatory mediators in microglia (Figure 6). NF-κB is a transcription factor found in all nucleated cell types. Studies have shown that it plays a vital role in cellular inflammation. Under normal physiological conditions, NF-κB is located in the cytoplasm. Once activated by LPS, the IκB subunit is degraded and phosphorylated, and the p65 subunit is transported into the nucleus, thereby regulating the production of pro-inflammatory mediators [47,48,49]. In this experiment, we found that IAL suppressed LPS-induced activation of NF-κB in BV2 cells (Figure 7). In Xing et al.’s study, IAL inhibited IKKβ kinase activity to interrupt the NF-κB/COX-2-mediated signaling cascade and induce apoptosis regulated by cofilin mitochondrial translocation in glioblastoma [50]. This showed that, with increasing doses, IAL can exhibit other pharmacological functions and other therapeutic effects on diseases. Heme oxygenase-1 (H0-1) is a key anti-inflammatory protein downstream of Nrf2 [51]. Studies have shown that IAL regulates the AKT/Nrf2/HO-1 pathway. In this experiment, we proved that IAL inhibited microglial inflammation by activating the AKT/Nrf2/HO-1 pathway (Figure 7). In a further investigation, we cultured SN4741 cells with supernatant of IAL and LPS-treated microglia. The results confirmed that IAL treatment protected SN4741 from microglial activation-mediated neurotoxicity (Figure 8).

## 5. Conclusions

Taken together, this study demonstrated that microglia inflammation and neuronal apoptosis played a vital role in a PD model, and IAL treatment ameliorated PD symptoms by inhibiting neuro-inflammation and neuronal apoptosis. Its specific molecular mechanism remains to be further studied. Our data suggest that neuro-inflammation and related-neuronal apoptosis may be a target for PD therapy, and IAL may become a candidate in PD treatment by targeting neuronal apoptosis.

## Figures and Tables

**Figure 1 cells-11-02927-f001:**
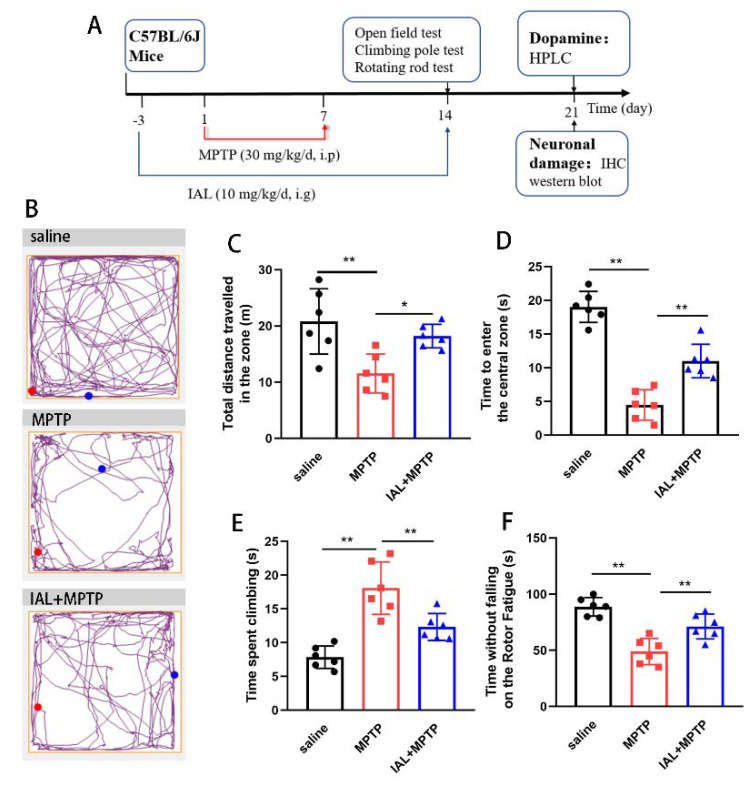
IAL alleviates the mobility of MPTP-induced PD mice. (**A**) Experimental process diagram. (**B**–**D**) The effect of IAL on the motor distance and time to enter the central area in MPTP-injected mice was tested via open field experiment. (**E**) The effect of IAL on the coordination ability of MPTP-injected mice was tested via a pole-climbing experiment. (**F**) The effect of IAL on the fatigue resistance of MPTP-injected mice was tested via rotating rod experiments. Results are shown as means ± SEM (n = 6). * *p* < 0.05, and ** *p* < 0.01 stand for biologically significant differences.

**Figure 2 cells-11-02927-f002:**
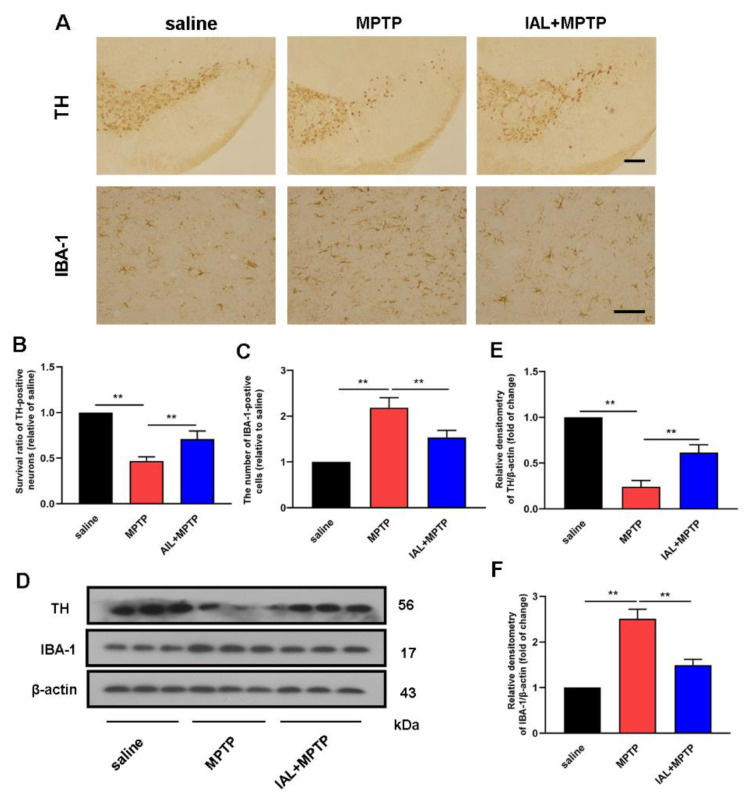
IAL decreases dopaminergic neuron degeneration and inhibits microglia over-activation in MPTP-induced PD mice. (**A**,**B**) The number of TH-positive cells in the SN was detected using the immunohistochemistry method (the bar is 50 μm). (**A**,**C**) The number of microglia in the SN was marked with IBA-1 using an immunohistochemistry method (the bar is 20 μm). (**D**–**F**) The proteins of TH and IBA-1 were detected using the Western blot method. Results are presented as means ± SEM (n = 3). In the figure, black is the control group, red is the MPTP group, and blue is the MPTP+IAL treatment group. ** *p* < 0.01 stands for biologically significant difference.

**Figure 3 cells-11-02927-f003:**
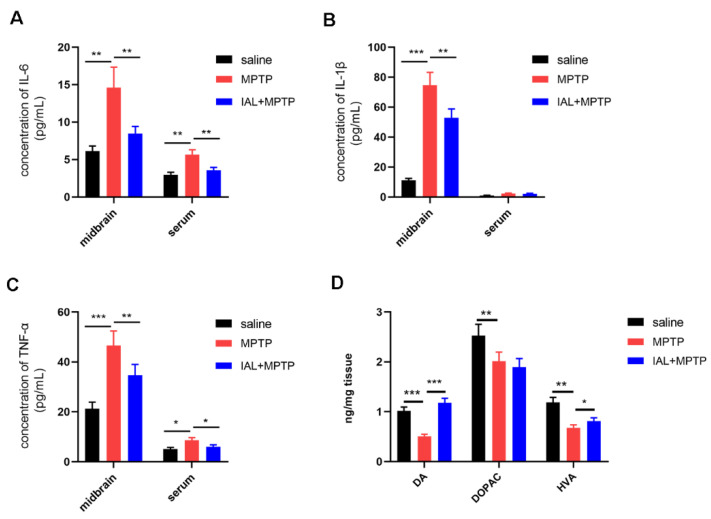
IAL suppresses midbrain inflammatory responses and increases striatal dopamine and its metabolite levels in MPTP-induced PD mice. (**A**–**C**) The expression of pro-inflammatory mediators (IL-6, IL-1β and TNF-α) in the midbrain and serum was detected with the ELISA method. (**D**) Levels of dopamine and its metabolites (DOPAC and HVA) in the striatum were detected by HPLC. Results are shown as means ± SEM (n = 3). * *p* < 0.05, ** *p* < 0.01 and *** *p* < 0.001 stand for biologically significant differences.

**Figure 4 cells-11-02927-f004:**
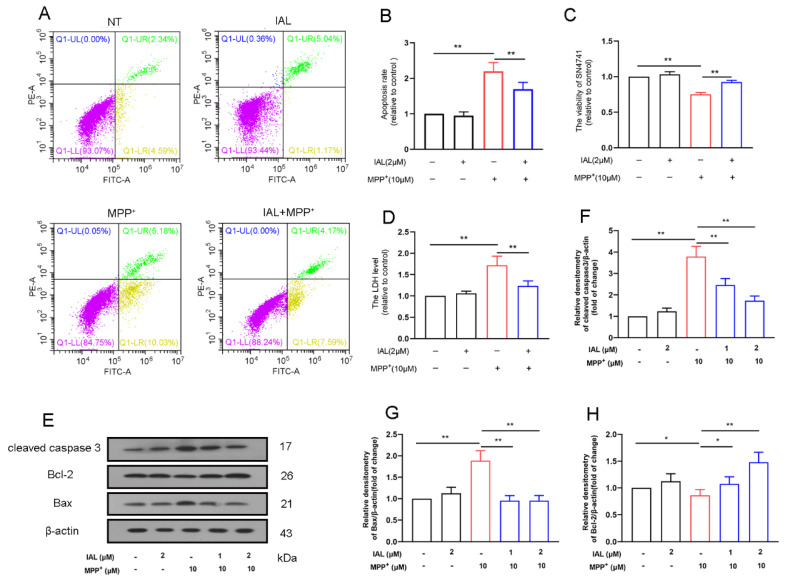
IAL reduces the cells death in MPP^+^-exposed SN4741 cells. (**A**,**B**) The effect of IAL on MPP^+^-induced SN4741 cell death was examined using flow cytometry. (**C**) The effect of IAL on cell viability was examined in MPP^+^-exposed SN4741 cells using the CCK-8 method. (**D**) The effect of IAL on cellular LDH release was examined in MPP^+^-exposed SN4741 cells using the LDH method. (**E**–**H**) The effect of IAL on apoptosis-related proteins in SN4741 was analyzed with the Western blot method. Results are presented as means ± SEM (n = 3). * *p* < 0.05 and ** *p* < 0.01 stand for biologically significant differences.

**Figure 5 cells-11-02927-f005:**
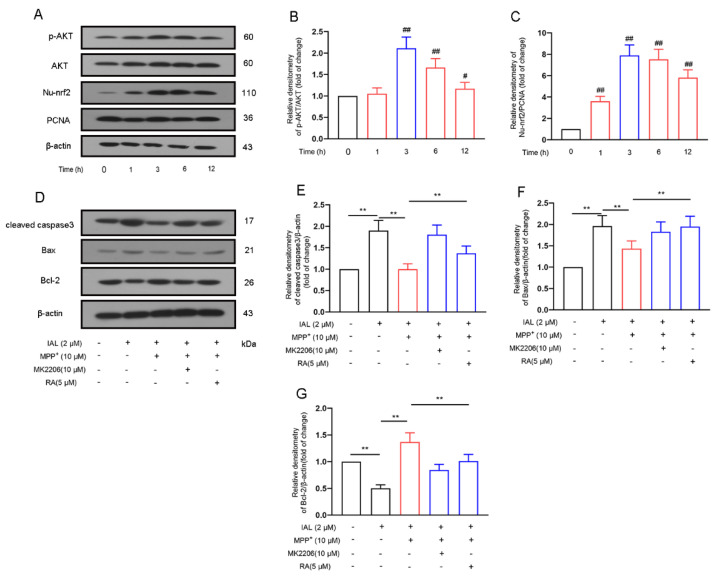
IAL treatment reduces MPP^+^-induced SN4741 apoptosis by activating AKT/Nrf2 pathway. (**A**–**C**) The effect of IAL on the AKT/Nrf2 pathway in SN4741 was analyzed with the Western blot method. (**D**–**F**) After pretreatment of cells with MK2206 and RA, the effect of IAL on apoptosis-related proteins in SN4741 was analyzed with the Western blot method. Results are shown as means ± SEM (n = 3). ^#^
*p* < 0.05, ^##^
*p* < 0.01, and ** *p* < 0.01 stand for biologically significant differences.

**Figure 6 cells-11-02927-f006:**
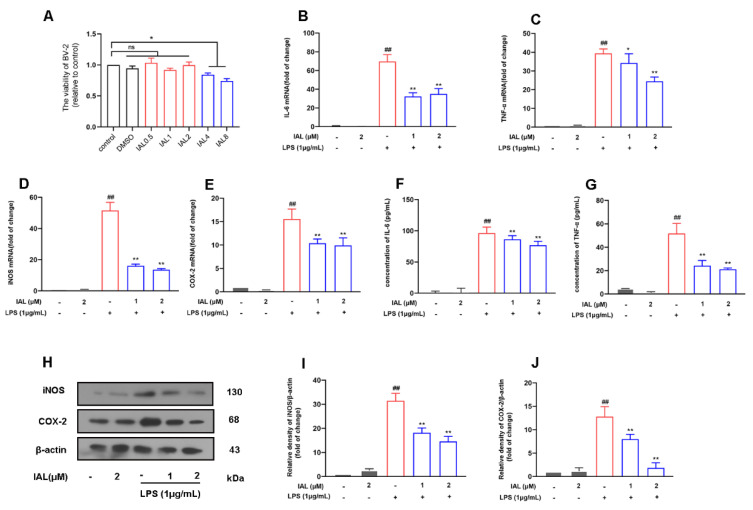
IAL inhibits the release of pro-inflammatory mediators in BV2 cells. (**A**) The effect of IAL on BV2 cell viability was examined with a CCK-8 assay. (**B**–**E**) The mRNA expression of the pro-inflammatory mediators IL-6, TNF-α, iNOS and COX2 was detected with a real-time PCR assay. (**F**,**G**) The protein expression of the pro-inflammatory mediators IL-6 and TNF-α was detected with an ELISA assay. (**H**–**J**) The protein expression of the pro-inflammatory mediators (iNOS and COX2) was detected with a Western blot assay. Results are shown as means ± SEM (n = 3). ns stands for no significant difference. ^##^
*p* < 0.01, * *p* < 0.05 and ** *p* < 0.01 stand for biologically significant differences.

**Figure 7 cells-11-02927-f007:**
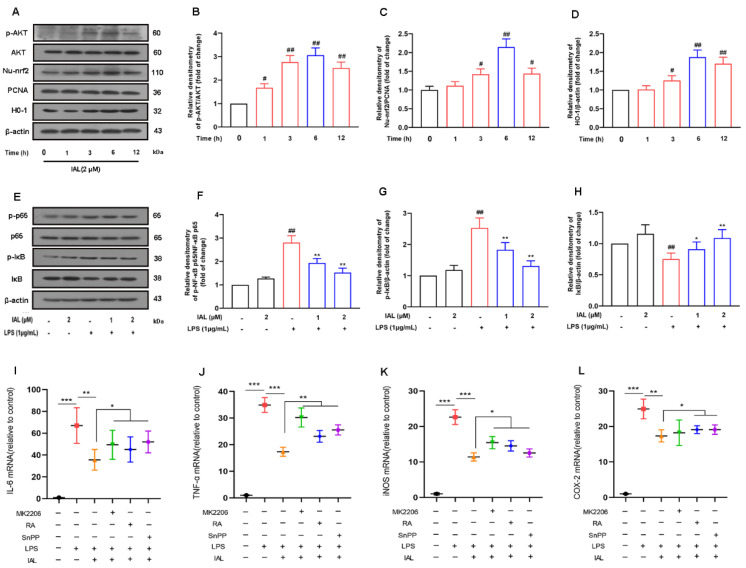
IAL treatment inhibits microglial inflammation by activating the AKT/Nrf2/HO-1 signaling pathway and inhibiting the NF-κB signaling pathway. (**A**–**D**) The effect of IAL on the AKT/Nrf2/HO-1 pathway in BV2 cells was examined with the Western blot method. (**E**–**H**) The effect of IAL on the NF-κB pathway in LPS-exposed BV2 cells was examined with the Western blot method. (**I**–**L**) After pretreatment of cells with MK2206, RA and SnPP, the effect of IAL on the mediators was examined using the real-time PCR method. Results are shown as means ± SEM (n = 3). ^#^
*p* < 0.05, ^##^
*p* < 0.01, * *p* < 0.05, ** *p* < 0.01 and *** *p* < 0.001 stand for biologically significant differences.

**Figure 8 cells-11-02927-f008:**
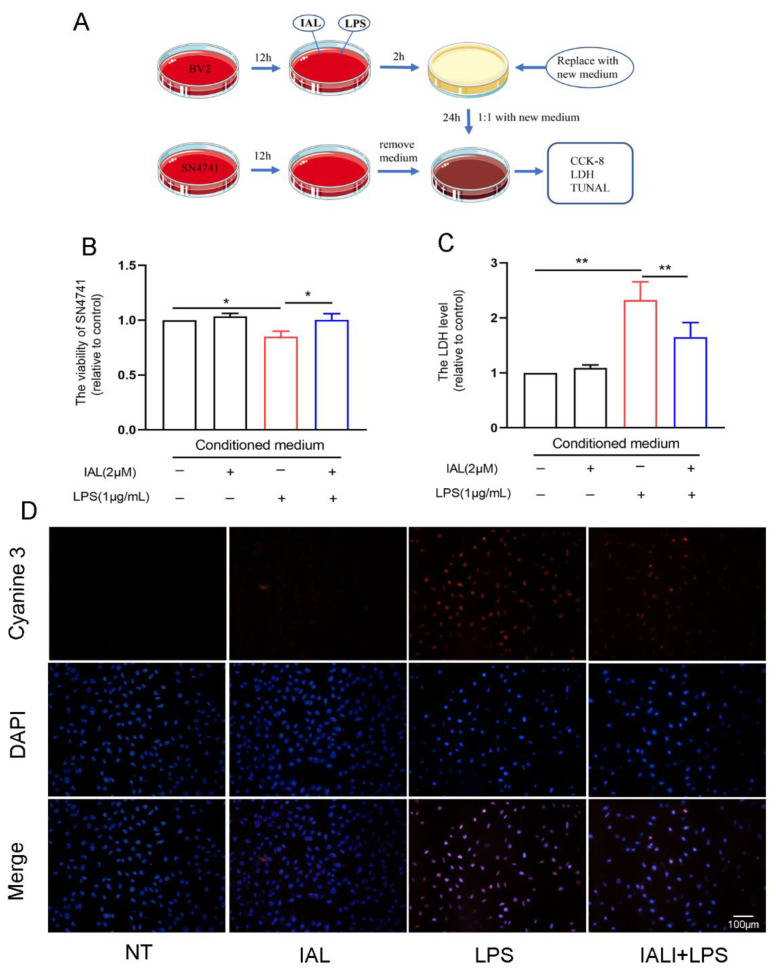
IAL treatment protects SN4741 from microglial activation-mediated neurotoxicity. (**A**) Conditioned medium preparation flow chart. (**B**) The effect of conditioned medium on SN4741 cell viability was examined using the CCK-8 method. (**C**) The effect of conditioned medium on cellular LDH release was examined using the LDH method. (**D**) The effect of conditioned medium on SN4741 cell death was examined using Tunel staining. Results are shown as means ± SEM (n = 3). * *p* < 0.05 and ** *p* < 0.01 stand for biologically significant differences.

**Figure 9 cells-11-02927-f009:**
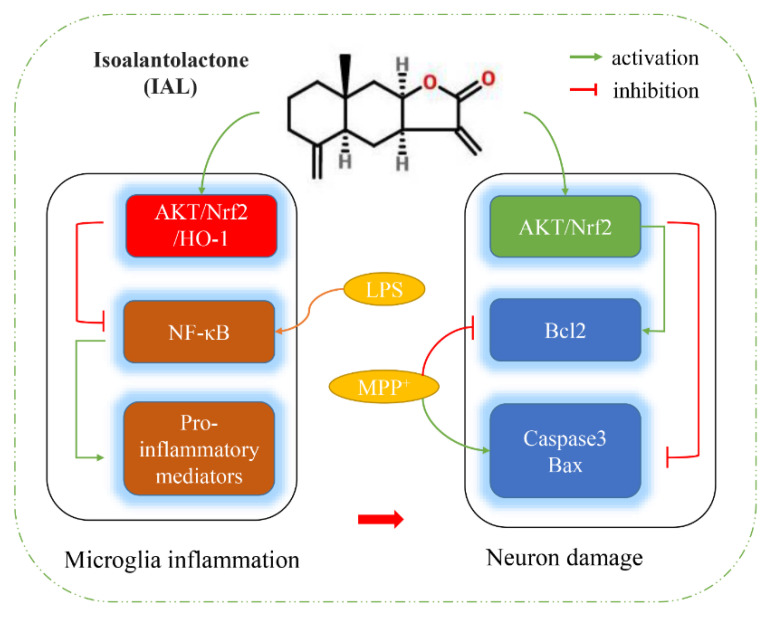
Flowchart of how IAL works. IAL regulates neuro-inflammation and neuronal apoptosis to ameliorate PD symptoms.

**Table 1 cells-11-02927-t001:** The primer sequence.

Gene	Sequences	Length (bp)
*IL-6*	(F) 5′-GACAAAGCCAGAGTCCTTCAGA-3′ (R) 5′-TGTGACTCCAGCTTATCTCTTGG-3′	76
*TNF-α*	(F) 5′-ACTGAACTTCGGGGTGATCG-3′ (R) 5′-TGGTGGTTTGTGAGTGTGAGG-3′	102
*iNOS*	(F) 5′-CAACAGGGAGAAAGCGCAAAA-3′ (R) 5′-TACTGTGGACGGGTCGATGT-3′	175
*COX-2*	(F) 5′-TGAGTACCGCAAACGCTTCT-3′ (R) 5′- CAGCCATTTCCTTCTCTCCTGT-3′	74
*β-actin*	(F) 5′-GTCAGGTCATCACTATCGGCAAT-3′ (R) 5′-AGAGGTCTTTACGGATGTCAACGT-3′	147

## Data Availability

All data are available in the manuscript.

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
