# Peer review of "Isoalantolactone (IAL) Regulates Neuro-Inflammation and Neuronal Apoptosis to Curb Pathology of Parkinson’s Disease"

_cells, 2022, doi:10.3390/cells11182927_

Round 1

Reviewer 1 Report

I believe this is a excellent experiment, but minor revision is needed.

Author Response

Thank you for your review and suggestions. Below we will reply one by one.

Reviewer 2 Report

Major Comments:

  1. The manuscript requires a good amount of rephrasing the sentences for a better clarity of message being conveyed, especially in Methodology.

  2. A quick Google search for the primer sequences used in the study showed that the exact same primer sequence has been used in another published study as a primer sequence for beta-actin. On running the sequence through Blast, I could not verify the authenticity and specificity of the primer sequences for COX-2 and GAPDH. Please consider this as a very serious query and an explanation for this discrepancy in primer sequences will be helpful. 

  3. Line 63: Please remove the word “patients” as the study in ref 16 was carried out in C57 mice.

  4. Line 359:“neuronal apoptosis to curb pathogenesis of PD”: The study does explore the potential of IAL against the molecular mechanisms, making such a definitive statement is inadvisable. The data and the study is not conclusive enough to state that IAL can curb pathogenesis of PD. Please rephrase.

Minor Comments:

  1. The manuscript requires a good amount of rephrasing the sentences for a better clarity of message being conveyed, especially in Methodology.

  2. Line 33: “Dopaminergic”

  3. The manuscript, in its current form, is not acceptable without extensive language editing. Although the quality and quantity of data in this manuscript is good, the language used makes it a little difficult and confusing to read and understand, at various instances.

Author Response

(The authors gave the same response as above.)

Reviewer 3 Report

Isoalantolactone (IAL) regulates neuro-inflammation and neuronal apoptosis to curb pathogenesis of Parkinson’s disease

The manuscript fundamental which emphasizes the use of IAL for the treatment of PD, is acceptable in the field of drug discovery. But to proceed further below points should be addressed.

1. How IAL dose 10ml/kg was selected for intra-peritoneal administration. Why single dose IAL was selected for research.

2. Why 10 mice chosen for each group (30 mice,3 groups), where as 6 mice would be sufficient for each group as per the animal ethical committee.

3. Why the activity of IAL was not interpreted with any standard control group.

4. One important pathological characteristics of PD is Parkinson’s disease dementia (PDD), which is due to neuroinflammation and neuronal apoptosis.(PMID: 33272323, 15535032). Parkinson's disease dementia is neuropathologically characterized by aggregates of α-synuclein (Lewy bodies) in limbic and neocortical areas of the brain. The dementia of the animal should be evaluated using any one model. 

5. Why open-field test performed?

Author Response

(The authors gave the same response as above.)

Reviewer 4 Report

The article was studied one of the most important issues in the modern world. This research is very important for human health. The scientific research done by the authors can be highly appreciated. But, it is still better to supplement the introduction section with the latest literature in this field. And it is not enough to discuss the results.

Author Response

(The authors gave the same response as above.)

Round 2

Reviewer 3 Report

This form of revised article can be accepted

Author Response

Thanks for your comment. Your review makes us better.